# Child Abuse: Adherence of Clinical Management to Guidelines for Diagnosis of Physical Maltreatment and Neglect in Emergency Settings

**DOI:** 10.3390/ijerph20065145

**Published:** 2023-03-15

**Authors:** Pierpaolo Di Lorenzo, Claudia Casella, Serena Dei Medici, Fabio Policino, Emanuele Capasso, Massimo Niola

**Affiliations:** Department of Advanced Biomedical Science-Legal Medicine Section, University of Naples “Federico II”, 80131 Naples, Italy; pierpaolo.dilorenzo@unina.it (P.D.L.); claudia.casella@unina.it (C.C.); fabio.policino@unina.it (F.P.); emanuele.capasso@unina.it (E.C.); masniola@unina.it (M.N.)

**Keywords:** child abuse, physical maltreatment, neglect, guidelines, violence

## Abstract

Child maltreatment is a phenomenon of great importance due to the significant socio-health implications related to it. Purpose of the study is assessing compliance child abuse clinical management with guidelines and suggest corrective actions to avoid false negative or false positive judgments. The data come from 34 medical records of child victims of suspected abuse hospitalized in a pediatric clinic. We examined diagnostic and medico-legal management through the analysis of pediatric, dermatological, ophthalmological (including fundus examination), and gynecological (only in some cases) consultations, brain and skeletal imaging, laboratory tests (with reference to the study of hemostasis), and medico-legal advice. Of 34 patients, the average age was 23 months, ranging from 1 month to 8 years. The judgment was positive for abuse for 20 patients and negative for 12 patients; in two cases it was not possible to express a conclusive judgment. Two children died because of the injuries sustained. We underline the need of clinical-diagnostic standardized protocols, coroner in emergency settings, short-distance follow-up, social worker support. We also suggest objectifying in a descriptive way (using a common and repeatable language) and with photographic documentation the results of all the investigations carried out, to evaluate signs of physical maltreatment and neglect.

## 1. Introduction

The World Health Organization offers the following definition of child abuse: “all types of physical and/or emotional ill-treatment, sexual abuse, neglect, negligence and commercial or other exploitation, which results in actual or potential harm to the child’s health, survival, development or dignity in the context of a relationship of responsibility, trust or power” [1].

In 2022, the World Health Organization also estimates that nearly three in four children—or 300 million children—aged 2–4 years regularly suffer physical punishment and/or psychological violence at the hands of parents and caregivers [1]. The deaths of child victims of abuse are underestimated because not all of them are reported, recognized, and proven [2,3]. The numbers of the phenomenon have increased significantly during the COVID-19 pandemic due to the considerable impact that the lockdown has had on the lifestyle and daily routine of most of the population [4]. Families have lived in physical isolation and without access to childcare because daycare centers, schools, and social services have been temporarily closed.

Child maltreatment remains a phenomenon of great importance due to the significant socio-health implications related to it. The main objective is prevention, through the highest alert among health workers to early identify signs of abuse. The responsibility of defining the signs and symptoms of child abuse falls on healthcare professionals at every point of care delivery, not just in emergency situations [5]. The diagnosis of child abuse is a delicate matter: in the event of an incorrect diagnosis, the child cannot benefit from early treatment. A study has reported the lack of knowledge among doctors on the subject of child abuse. It has been found that almost 78% of medical students have little knowledge of this topic, which obviously also affects their preparation in their subsequent professional life [6].

The multiplicity of injuries indicative of maltreatment requires a multi-professional (healthcare and non-healthcare personnel) and multidisciplinary approach that cannot escape from medico-legal advice and from a good level of coordination and collaboration between the different areas of relevance [7]. The management of child abuse in pediatric emergency departments is difficult due to the complexity of screening and lengthy care processes and necessarily requires a specialized hospital team both in the initial phase of assistance and in the subsequent monitoring of abuse victims [8]. More than all orthopedic injuries indicate the suspicion of abuse, suggesting that the doctors proceed with further investigations or interventions [9,10].

Currently, there are no standardized protocols for ascertaining cases of maltreatment. An interesting systematic review [11] comparing twenty national or regional guidelines [12,13,14,15,16,17,18,19,20,21,22,23,24,25,26,27,28,29,30,31] issued by academic societies or health agencies in high-income countries from 2010 to 2020 on early detection and diagnostic workup in child physical abuse. This review identified guidelines incompleteness in the management and follow-up of wounds and discrepancies in diagnostic processes. Guidelines have shown contradictions in the terminology, for example the concept of sentinel injury is not shared by all experts. Six guidelines [12,16,19,23,24,31] include only cutaneous sentinel injuries (hematomas, bruises, burns, abrasions, lacerations, and scars), in others [13,14,15,20,21,22,24,27,28,30] skeletal, intraoral, intracranial, or abdominal injuries. Guidelines are heterogeneous on the tests to be performed in ascertaining signs of abuse. The radiological study of the skeleton is always systematically recommended for the purpose of detecting the occult lesion. Review has shown that in 70% of the guidelines the radiological follow-up is mentioned (recommended between 7 days and 3 weeks after the first investigation; with an average of 14 days for most of the guidelines), to be performed systematically or only in case of persistence of diagnostic doubt. Only half of the guidelines [12,13,14,15,16,18,19,20,21,22,23,24,28,30,31] recommend the study of bone metabolism, through serum calcium, phosphorus, and alkaline phosphatase (systematically or case by case) or calcium alone. In seven guidelines [12,14,16,22,24,27,28], testing for parathyroid hormone and 25-hydroxy-vitamin D is indicated either systematically or according to the clinical context.

Three guidelines [22,27,28] recommend fibroblast culture and/or DNA analysis in osteogenesis imperfecta suspecting. Finally, bone scintigraphy is mentioned only in 70% of the guidelines (14 out of 20); in five guidelines [14,16,22,24,28], the serum dosage of copper and ceruloplasmin is recommended, not always in a systematic way. Fundus examination is recommended in sixteen guidelines (systematically or based on the clinical context). Only five guidelines [13,16,18,21,22] pose an upper age limit to perform fundus, ranging from 1 year to 5 years. All the guidelines reviewed recommend a head CT (computer tomography), either systematically or to be evaluated on a case-by-case. The use of contrast agent, although mentioned in seven guidelines, is never recommended. Brain MRI (magnetic resonance imaging) is recommended in 95% of cases (19 out of 20 guidelines), although in some guidelines only when CT has detected abnormal results.

In two guidelines [15,31], both CT and MRI could be performed systematically, regardless of the initial CT results or to choose one of the two tests.

Although the need to explore primary hemostasis and coagulation is shared, there are obvious discrepancies in the tests to be performed. Platelet evaluation is systematically recommended in six guidelines [14,15,16,23,24,28]; in others, depending on the clinical context [12,13,18,20,21,22,27,31]; in still others [13,27], only in the presence of bruises. Three guidelines [24,27,28] recommend specific coagulation tests; others [12,15,16,20,21,22,31] do not mention the type of test. A total of 7 out of 20 guidelines [15,16,20,21,22,24,28] recommend advanced coagulation tests either systematically or depending on the clinical context.

Therefore, our study objective was to analyze clinical management in 34 cases of suspected physical child maltreatment (treated by doctors in pediatric hospital) through semeiotics of the abuse, clinical-forensic differential diagnosis and correct collection of the highlighted signs. This to assess compliance with guidelines and suggest corrective actions to avoid false negative or false positive judgments. A high number of false positives and negatives are especially evident when screening tools for potential child maltreatment are used [32,33]. Studies on the misdiagnosis of child abuse show, for example, that bruises are the most common injuries, but also those that are most frequently overlooked or diagnosed as non-violent. Failure to recognize bruising caused by physical child abuse represents errors in medical decision that directly contribute to poor patient outcomes. The TEN rule (identification of suggestive bruising on torso, ear, and neck) does not allow 19% of abused patients to be recognized [32,34].

Thus, our purpose is in view of child protection and univocal management in the event of suspected physical abuse.

## 2. Materials and Methods

The data used for this study were extrapolated from medical records of children victims of suspected abuse treated in a pediatric hospital from July 2017 to August 2022.

The initial sample consisted of medical records of 41 patients, among which, for the purposes of our study, 34 were selected, excluding 7 patients for which was formulated specific suspicion of sexual violence. We have included all forms of physical abuse and neglect, instead we have excluded psychic abuse and sexual violence.

We analyzed diagnostic and medico-legal management.

For each small patient, the medical documentation, drawn up by the health professionals, was assessed, including anamnestic, clinical, instrumental, and laboratory data were collected: pediatric, dermatological, ophthalmological (including fundus examination), otorhinolaryngological, neuropsychiatric, rheumatological, neurosurgical and gynecological (only in some cases) consultations, brain and skeletal imaging, laboratory tests (with reference to the study of hemostasis), and medico-legal advice to give an opinion on the compatibility of the injuries with a traumatic origin, accidental or abusive. A total of 30 patients were also subjected to a medico-legal examination, even in the presence of a social worker.

From the analysis of each medical record, the following elements were identified: age; sex; circumstantial and anamnestic elements useful for the purposes of abuse compatibility; motivation for emergency observation; objective clinical findings in the emergency room; outcome of laboratory investigations and specialist consultations carried out during hospitalization; findings from diagnostic imaging investigations; behavior and psychological status of the child, additional findings from medico-legal observation and timing of its execution; conclusions about the compatibility of the lesions; and final orientation on hypothesis of maltreatment.

The moments of evaluation were survey, identification, and objectivation of signs and clinical manifestations, differential diagnosis between clinical signs and manifestations of traumatic and/or pathological etiology, differential diagnosis between accidental and non-accidental traumatic injury, differential diagnosis between natural and induced pathological lesion, evaluation of the hypothesis of a crime. The observation was carried out either in the emergency room or during the hospital stay.

During the anamnesis, attention was paid to eating habits, sleep-wake, alvus, diuresis, and any previous accesses to the emergency room or hospital admissions. Data relating to previous medical observations in the same hospital were captured through the consultation of the hospital’s internal computer database and the data relating to hospitalizations carried out in other structures through paper medical records and reports presented by child’s parents.

The general conditions were noted: nutrition, hydration, and any signs of neglect (hair, teeth, hygiene, and absence of compulsory vaccinations).

The whole body, sometimes including the mouth, was examined to highlight any recent and/or previous lesions. Lesions description was detailed about topographical location and morphochromatic characteristics.

Laboratory tests were performed to evaluate blood counts, electrolytes, liver and kidney function indices, hemostasis, bone metabolism, psychotropic substance intake, and genetic tests.

The instrumental investigations used were total body and segmental radiographs, CT and MRI of the skull and spine, fundus examination, and abdominal ultrasound.

## 3. Results

Before showing our results, we specify that weaknesses in the study are the small number of cases and the need to broaden the assessment, not only with respect to the number of cases, but also with respect to the health structures involved, hoping for the future, since the deficiencies highlighted, to develop a univocal management protocol of the child abuse shared by all hospital facilities. The aim is to avoid cases of misdiagnosis with the aim of ensuring the protection of the abused subjects.

Medical records were about 34 patients, 17 males and 17 females, hospitalized from July 2017 to August 2022; the average age was 23 months, ranging from 1 month to 8 years (Figure 1).

In the first six months of life, acts of violence are more frequent in male subjects (10 males out of a total of 14 abused subjects); after this chronological limit, data is reversed.

For each case, the type of lesions and the clinical, laboratory and instrumental tests performed were identified (Table 1).

The psychic state was fear, feeling of helplessness and/or horror, detachment, absence of emotional activity, feeling of lightheadedness, dissociative amnesia, and inability to remember important aspects of the trauma.

Type of injuries were erythema, bruising, hematoma, abrasions/excoriations, continuous solutions, lacerated-contused wounds, scars, fractures, and previous fractures; morphology of injuries were scratches, bites, manual grasping, and burns (e.g., from cigarettes and forced immersion) to attributability to the means of production. Injures chronology were ascertained: finding of lesions in different developmental phases (bruises and hematomas with different chromatic evolution, solutions continuously bleeding or under scab and scars, fractures, and bony calluses).

For 20 patients, the judgment was conclusive for a hypothesis of abuse, in 12 cases this hypothesis was excluded. In two cases it was not possible to express a conclusive judgment. Two children died as a result of the injuries sustained, one of these cases was initiated in a judicial process (Figure 2).

Of the positive cases, two assessments were accompanied by the recommendation of a short-term follow-up in order not only to carry out a strictly clinical evaluation, but to investigate the appearance of further signs of maltreatment. A judgment was formulated with a recommendation to study in deep coagulation tests to rule out genetic and/or acquired hemostasis diseases.

Among the negative cases, a psychological exploration of the mother was recommended.

In one of the two cases with an inconclusive judgment, it was the firstborn of twins conceived using the in vitro fertilization method [35].

Only one child was observed on two occasions three months apart, for two distinct traumatic events.

Only 9 children had a medical history suggestive for traumatic events that occurred in the 24 h prior to the observation. Three children were accompanied by police officers.

In the 30 patients in which direct forensic observation was carried out, the time elapsed from hospital admission was variable from 0 to 40 days, with an average of 10 days.

We detected 6 cases of neglect, among which a 4-year-old child presented a worsening underlying disease (systemic sclerosis) due to parents’ little attention to care. In another case, it was necessary to deepen the action of sodium valproate (which the minor took for the treatment of West syndrome) on bone metabolism [36], considering the multiple fractures found in various developmental phases. This child also presented positivity for taking psychoactive substances.

In another case, dermatological evaluation and skin biopsies were required for diagnosis of gangrenous pyoderma, excluding traumatic origin of scalp injuries. In one more child, the underlying vasculitis lesions were appropriately evaluated to exclude traumatic injuries.

In three cases, the dynamics reported by the parents were not compatible with the expected modalities of production of the injuries. In seven cases, the reported dynamics were compatible with vigorous shaking in infants aged 3 to 6 months.

Coagulation tests were difficult to interpret, especially in cases where the traumatic trigger factor was decisive.

In three cases, it reported unblocking maneuvers of parents, following cyanosis after inhalation of food, or the execution of resuscitation maneuvers, suggested by telephone by the 118 health professionals. Injuries in these cases were compatible with the trauma caused by energetic shaking.

In the case of a 4-month-old child, the reported reason for access was a polytrauma from a road accident. The injuries, consisting of head trauma and pulmonary and hepatic contusions, hematoma in the right zygomatic and in the right inguinal regions, were compatible with the outcomes of road trauma. However, a short-term follow-up was suggested in order to protect the minor in relation to social context and an unquiet family environment due to frequent arguments between parents.

## 4. Discussion

The data analysis, in accordance with the study of the literature, allowed the detection of a series of criticalities in the clinical approach in cases of suspected child maltreatment. These criticalities have repercussions on the medico-legal evaluation, which encounters, in the absence of a complete clinical and instrumental study, many difficulties in providing a conclusive opinion about the need or not to protect the child.

Moreover, an incorrect diagnosis could lead to further injuries or even child death or affect the action of the Judicial Authority [37].

Therefore, early diagnosis of child maltreatment, based on sentinel lesion recognition and careful clinical evaluation, imaging, and laboratory tests, are essential.

To assist healthcare professionals in identifying child maltreatment, several academic societies across the world have developed guidelines. However, despite these standardization efforts, several studies have reported suboptimal adherence to them in health care. For example, in 2018, 36% of doctors in four European countries felt that a 10-week-old newborn with oral bleeding was not considered at risk for maltreatment [38]. From a French survey conducted in 2015 [39] emerged that only 28% of pediatricians would have prescribed MRI of the head as part of the diagnostic process for a suspected abuse in a 9-month-old child with a fractured head, femur, multiple bruises, and head trauma.

In the cases we examined, there was often a lack of adequate collection not only of the anamnestic data, but also of the circumstantial elements, which in the remaining cases, were found to be diriment.

From our analysis in a minority of cases (only 4) the coroner observed the child immediately after hospitalization or at most the following day. In all other cases, the examination was taken place only after several days, clearly losing valid elements for the assessment. In most cases, the coroner did not meet child caregiver. Other times he was called during hospitalization, when the healthcare professionals were alarmed by the deteriorating evolution of the clinical status. The hospital structure where children were hospitalized does not provide for a coroner expert in child abuse, but makes use of external consultants. In one case, the coroner was alerted only 20 days after the observation in the emergency room when during the hospitalization a bone re-fracture was highlighted. This child on admission the child presented a previous femoral fracture occurred about three months earlier, and there was a report to the social services of neglect by the mother, confirmed by the finding of poor hygiene with dermatitis and widespread scratching lesions on the physical examination of the genitals.

Our analysis found that the signs of neglect, which often accompany cases of maltreatment, tend to be ignored in clinical evaluation. For example, the exploration of the oral cavity was made almost exclusively during medical-legal observation. In one case, the need for dental advice was highlighted. Instead, dental valuation is never included in the protocols for the management of child abuse. Among medical records analyzed, it was a two-year-old child with multiple facial and mucosal excoriations of the lip and dental fractures affecting the upper central incisors. In this case, dental consultancy would have been necessary to also establish the contemporaneity of the dental lesions with respect to the facial and oral ones considering the hypothesis of an injurious recurrence.

In the cases examined, the study of hemostasis and fundus examination was always performed, suggesting their standardization. In this regard, it must be noted that the ocular lesions can be different and there is not a characteristic lesion. Retinal hemorrhages are most often a sign of significant trauma, and the severity of the hemorrhages is generally proportional to the severity of the neurological trauma, despite the difficulty in differentiating accidental from inflicted trauma [28]. However, in our cases the ophthalmological evaluation was never accompanied by photographic material, but only by the descriptive report. This represents a limitation as the objectification of the lesions allows their definition and demonstration in the subsequent possible forensic context. We emphasized the need to always produce all photographic documentation and report and preserve all useful elements, to allow a re-evaluation over time.

The same criticality occurs with the description of external mechanical damage, in the adjective of erythema, ecchymosis, hematomas, abrasions, excoriations, continuous solutions, and lacerated-contused wounds. Not infrequently it happens that the same lesions are objectified in a different way by the clinicians, thus modifying the possible production mechanism, and therefore, the compatibility with the referred dynamics. Again, the lack of specific references about the size of the injuries, which only photographic documentation and objective measurement can resolve.

In our cases, very often the forensic examination occurred several days after the child access to the emergency room. Thus, the lesions initially described by the clinicians appeared modified and, in the absence of photographic findings, the fundamental information that could derive from them were no longer useful in formulating conclusive judgment.

The study of bone metabolism was not routinary. In a patient, it was suggested considering of bone re-fracture during hospitalization.

Finally, psychological evaluation of the parents never performed (only in one case out of 34 a neuropsychiatric evaluation of the mother was executed). It can be useful for the purposes of the final evaluation. In one case, for example, despite a negative opinion on the hypothesis of abuse, a psychological exploration of the parents was recommended. In this phase, the support of social workers is useful to report on the socio-economic-cultural context of origin.

The study of the literature does not provide unique indications about the imaging and laboratory tests to be performed, whether they (and which of them) should be performed systematically or not. In our cases, the radiological study of the skeleton has always been performed except for a purely dermatological case in the absence of suspicion for bone lesions. This is in accordance with the scientific literature [11] as the radiological study of the skeleton is systematically recommended in 17 out of 20 guidelines with the aim of detecting occult skeletal lesions; in the remaining 3 guidelines, it is recommended on a case-by-case. The follow-up skeleton radiological investigation is also mentioned in 70% of the guidelines. However, the literature highlights the complexity of dating fractures from a radiological point of view [40].

The study of bone metabolism has never been carried out routinely, and in one case, it was required as a solution to the etiology of a fracture that occurred during hospitalization. However, this differs from the indications of the guidelines, of which, however, only half recommend the study of metabolism, through systematic serum dosage of calcium, phosphorus and alkaline phosphatase.

The fundus examination was carried out in 25 patients, however excluding the cases in which cranial and facial injuries were absent. This is in line with the findings of the scientific literature that it is unlikely to find significant retinal hemorrhages without head trauma and with negative neuroimaging; therefore, a routine retinal examination may not be required in such cases [41]. The child age was not indicative for the choice to carry out the fundus examination. In 16 guidelines, the fundus examination is recommended, systematically or considering the clinical context; only in 5 international guidelines an upper age limit to perform it was, ranging from 1 year to 5 years.

Cranial and cerebral imaging was carried out in all our cases because neurological symptoms or head injury were often the reasons for access to treatment [42]. This is in accordance with the guidelines that recommend a head CT, either systematically or to be evaluated on a case-by-case. In fact, the number of occult head injuries is very high, especially in children with the following risk factors: rib fractures, multiple fractures, and facial injuries, age < 6 months. Therefore, even in the absence of neurological symptoms, screening with CT or MRI is recommended [43].

The coagulation study was performed in 14 out of 34 cases, sometimes in relation to reported anamnestic genetic predispositions, with reference to PT, Anti-Thrombin III, Von Willebrand Factor Antigen, VIII Factor, D-Dimer, and XIII Factor. Some guidelines recommend specific tests (activated partial thromboplastin time, prothrombin time, international normalized ratio and fibrinogen) or systematically or in case of contusions; others do not mention the type of test. A total of 7 out of 20 guidelines recommend advanced coagulation tests (dosage of factors VIII and IX, and von Willebrand activity with or without factor XI and XIII levels) systematically or depending on the clinical context.

We found that current guidelines formulate recommendations that leave choice on a case-by-case. However, this autonomy can lead to unreliable results. False negatives expose children to an estimated risk of relapse of 35% to 50% [44,45], associated with short- and long-term morbidity and mortality. In addition, false positives can delay the diagnosis of a serious underlying disease, such as bone fragility (e.g., osteogenesis imperfecta) or bleeding disorders (e.g., hemophilia) and lead to an inappropriate decision to protect the minor [11,28].

If there is a suspicion of child physical abuse, the clinician should determine whether the reported injury is compatible with the history provided by the caregiver. While it is often obvious that an injury is the result of maltreatment, it is difficult to say when it was inflicted and by whom. Timing is often a challenge as medical science cannot be highly specific [46].

A bleeding on brain CT can be classified as “acute”, however, it covers a temporal range that goes from hours to a few days. In this period, the child may have been under the protection of more than one person. Furthermore, the child, depending on age, often cannot provide his own story because he is not yet able to speak, or is too sick or scared. The anamnestic story is therefore entrusted entirely to the child’s caregiver, who can be the same person suspected of being responsible for the maltreatment.

## 5. Conclusions

The alarming numbers of child abuse request maximum alert among health professionals to highlight the signs earlier. From the point of view of the healthcare professional, a standardization of clinical-diagnostic protocols is essential. Currently, this standardization has still not been achieved both due to the variability of the guidelines on the subject and due to poor adherence to them.

Our analysis highlights the need for the figure of the coroner, in emergency contexts, with availability H24 or at least during the day, or on remote consultancy, thanks to the tools that the current technological advancement provides. We believe that the clinical observation is not enough. Only the observation conducted by the coroner, through injuries objectification and anamnestic and circumstantial collection is useful for compatibility with the reported dynamics.

It is necessary to objectify not only in a descriptive way, but also with photographic documentation the results of all the investigations carried out (for example fundus, external harm) to allow immediate evaluation of the case and its discussion in court. In this regard, it is also suggested the need, in case of absence of the coroner from the first observation, that the pediatrician evaluate some signs of neglect that are often ignored, in oral cavity (multiple caries, poor hygiene, dental fractures, or avulsion) and genitals (hygiene, and untreated advanced stage diaper rash).

A dental consultation would also be useful for the dating of dental lesions in a context of facial trauma.

We suggest providing for short-distance follow-up, especially in doubtful cases, to follow the child over time and highlight any signs of abuse recurrence. It is always useful to provide support from a social worker who can report about the socio-economic-cultural context. We propose a psychological interview for parents (or caregivers), which very often is not carried out except in cases of incompatibility or contradiction of the referred dynamics. We also suggest training for parents in terms of pediatric unblocking maneuvers, which have often been shown to be probably the cause of the shaking injury.

While coagulation test is routine in the hypothesis of abuse, the study of bone metabolism is almost never carried out.

Routinely radiological study of the skeleton in cases of weak clinical suspicion remains problematic due to the principle of caution, which suggests child protecting from unreasonable exposure to ionizing radiation.

In conclusion, we suggest training courses for health personnel to use a common language to describe the external harm of abuse, to know the guidelines and to provide a detailed clinical analysis useful to identify, objectify, and document the signs of maltreatment in the direction of the immediate protection of the child. Due to the delicacy and social importance of the topic, it is desirable to invest more resources in the training of health professionals, starting from university studies. All this is in order to improve knowledge on the subject and allow early identification and early diagnosis of child abuse. In the same way, parental training is also desirable to prevent and reduce the recurrence of the phenomenon.

## Figures and Tables

**Figure 1 ijerph-20-05145-f001:**
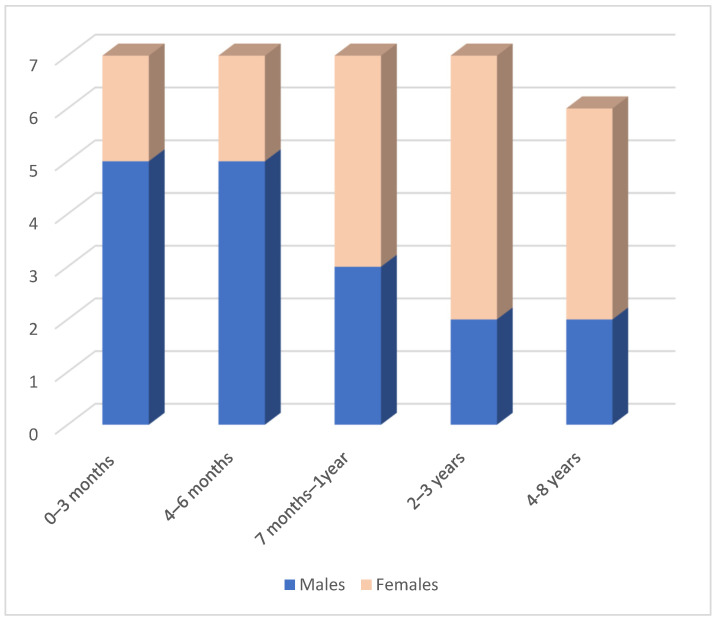
Sample distribution by sex and age.

**Figure 2 ijerph-20-05145-f002:**
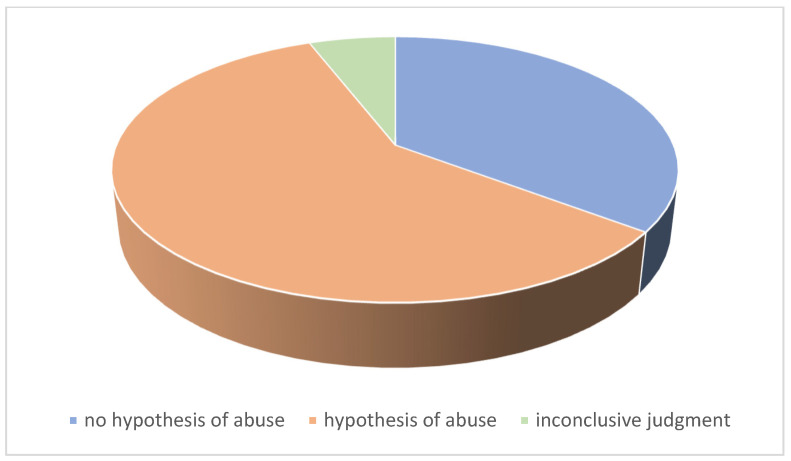
Conclusive medico-legal judgment.

**Table 1 ijerph-20-05145-t001:** Type of objective injury and the clinical, laboratory, and instrumental tests performed.

			# 1	# 2	# 3	# 4	# 5	# 6	# 7	# 8
age			5 m	9 m	4 m	1 m	4 m	8 m	3 m	2 y
sex			M	M	M	M	F	M	M	M
observed lesions	integumentary		+	+	-	+	+	-	+	+
	dental		-	-	-	-	-	-	-	-
	neglect		-	+	-	-	-	-	-	-
	skeleton		+/Nr	+/Pr	+/Nr	+/Pr	+/Nr	+/Nr	+/Nr	-
	fracture sites			5c; 3p		2p				
instrumental investigations carried out	neuroradiology (CT and MRI)		+/Pr	+/Pr	+/Pr	-	+/Pr	+/Pr	+/Pr	+/Pr
	fundus		+/Pr	-	+/Pr	+/Pr	+/Pr	+/Pr	+/Pr	-
	abdominal ultrasound		-	-	-	-	+/Pr	-	-	-
laboratory investigations carried out	coagulation TEST		+/Nr	-	+/Nr	+/Nr	+/Nr	+/Pr	-	+/Nr
	bone metabolism		-	-	-	-	-	-	-	-
	genetic tests		+/Pr	-	-	-	-	+/Nr	-	-
	laboratory (complete blood count, electrolytes, liver and kidney function)		+/Pr	+/Nr	+/Nr	+/Nr	+/Nr	+/Pr	+/Pr	+/Nr
	toxicology		-	-	-	-	-	-	-	+/Nr
carried out consultations	neuropsychiatric		-	-	-	-	-	-	-	-
	rheumatology		-	-	-	-	-	-	-	-
	neurosurgical		+/Pr	-	-	-	+/Pr	-	+/Pr	-
	dermatological		-	-	-	-	-	-	-	-
	orl		-	-	-	-	-	-	-	-
	gynecological		-	-	-	-	-	-	-	-
	pediatric		+	+	+	+	+	+	+	+
	ophthalmological		+	-	+	+	+	+	+	-
	medico-legal		-	+	-	+	+	+	-	+
			**# 9**	**# 10**	**# 11**	**# 12**	**# 13**	**# 14**	**# 15**	**# 16**
age			8 y	3 m	7 m	6 y	6 m	1 y	2 y	3 y
sex			F	F	F	M	F	F	F	F
observed lesions	integumentary		+	+	-	+	-	+	+	+
	dental		+	-	-	-	-	-	-	-
	neglect		+	-	-	-	-	-	+	-
	skeleton		+/Nr	+/Nr	+/Pr	+/Nr	+/Nr	+/Pr	+/Pr	+/Nr
	fracture sites				2c; 1p			1p	1p	
instrumental investigations carried out	neuroradiology (CT and MRI)		+/Pr	+/Pr	+/Nr	+/Pr	+/Pr	-	+/Pr	+/Pr
	fundus		-	+/Nr	+/Nr	+/Nr	+/Pr	+/Nr	+/Pr	-
	abdominal ultrasound		+/Nr	-	+/Nr	-	-	+/Nr	-	-
laboratory investigations carried out	coagulation test		+/Nr	+/Pr	+/Nr	-	+/Pr	+/Pr	-	-
	bone metabolism		-	-	-	-	-	-	+/Nr	-
	genetic tests		-	-	-	-	-	-	-	-
	laboratory (complete blood count, electrolytes, liver and kidney function)		+/Nr	+/Nr	+/Nr	+/Nr	+/Nr	+/Nr	+/Nr	+/Nr
	toxicology		-	-	-	-	-	-	-	-
carried out consultations	neuropsychiatric		-	-	-	+/Pr	-	-	-	-
	rheumatology		-	-	-	-	-	-	-	-
	neurosurgical		-	-	-	-	-	-	-	-
	dermatological		-	-	-	-	-	-	+/Pr	-
	orl		+/Pr	-	-	-	-	-	-	-
	gynecological		-	-	-	-	-	-	-	-
	pediatric		+	+	+	+	+	+	+	+
	ophthalmological		-	+	+	+	+	+	+	-
	medico-legal		+	+	+	+	+	+	+	-
		# 17	# 18	# 19	# 20	# 21	# 22	# 23	# 24	# 25
age		2 y	1 y	2 y	6 m	8 y	2 m	4 y	3 m	5 m
sex		M	M	F	M	M	M	F	F	M
observed lesions	integumentary	+	+	+	-	+	+	+	-	-
	dental	+	-	-	-	-	-	-	-	-
	neglect	-	+	-	-	-	-	-	-	-
	skeleton	+/Nr	-	-	+/Nr	+/Nr	+/Nr	+/Nr	+/Nr	+/Pr
	fracture sites									2c
instrumental investigations carried out	neuroradiology (CT and MRI)	+/Pr	-	-	+/Pr	+/Pr	+/Pr	-	+/Pr	+/Pr
	fundus	+/Nr	+/Nr	-	+/Pr	+/Pr	+/Nr	-	+/Pr	+/Pr
	abdominal ultrasound	-	+/Nr	+/Nr	-	-	-	+/Nr	-	-
laboratory investigations carried out	coagulation test	+/Nr	+/Nr	+/Nr	+/Pr	+/Pr	+/Nr	+/Nr	+/Pr	+/Nr
	bone metabolism	-	-	+/Pr	-	-	-	+/Pr	-	-
	genetic tests	-	-	-	-	-	+/Nr	-	-	-
	laboratory (complete blood count, electrolytes, liver and kidney function)	+/Nr	+/Nr	+/Nr	+/Pr	+/Nr	+/Nr	+/Nr	+/Nr	+/Nr
	toxicology	-	-	-	-	-	-	-	-	-
carried out consultations	neuropsychiatric	-	-	-	-	-	-	+/Nr	-	-
	rheumatology	-	-	-	-	-	-	-	-	-
	neurosurgical	-	-	-	-	+/Pr	-	-	-	-
	dermatological	-	+/Pr	+/Pr	-	-	-	+/Pr	-	-
	orl	-	-	-	-	-	-	-	-	-
	gynecological	-	-	-	-	-	-	+/Nr	-	-
	pediatric	+	+	+	+	+	+	+	+	+
	ophthalmological	+	+	-	+	+	+	-	+	+
	medico-legal	+	+	+	+	+	+	+	+	+
		# 26	# 27	# 28	# 29	# 30	# 31	# 32	# 33	# 34
age		8 y	4 y	1 m	20 d	8 m	2 y 3 m	5 m	3 y	8 m
sex		F	F	M	M	F	F	M	F	F
observed lesions	integumentary	+	+	-	+	+	+	-	+	+
	dental	-	-	-	-	-	-	-	-	-
	neglect	+	+	-	-	-	-	-	-	-
	skeleton	+/Pr	+/Pr	+/Pr	+/Pr	/Nr	+/Nr	+/Pr	+/Nr	-
	fracture sites	6c	1p	2c	1c; 4p			4c; 5p		
instrumental investigations carried out	neuroradiology (CT and MRI)	-	+/Nr	-	+/Nr	-	-	+/Pr	+/Nr	-
	fundus	+/Nr	+/Pr	+/Nr	+/Nr	+/Nr	+/Nr	+/Pr	+/Nr	-
	abdominal ultrasound	-	+/Nr	-	+/Nr	-	-	+/Pr	-	+/Pr
laboratory investigations carried out	coagulation test	+/Nr	-	-	+/Nr	-				
	bone metabolism	+/Pr	-	-	-	-	-	+/Pr	-	+/Pr
	genetic tests	-	-	+/Nr	-	-	-	-	-	-
	laboratory (complete blood count, electrolytes, liver and kidney function)	+/Pr	+/Nr	+/Nr	+/Nr	+/Nr	-	-	-	-
	toxicology	+/Pr	-	-	-	-	+/Nr	+/Nr	+/Nr	+/Nr
carried out consultations	neuropsychiatric	-	-	-	-	-	-	-	-	-
	rheumatology	-	-	-	-	-	-	-	+/Pr	+/Pr
	neurosurgical	-	-	-	+/Nr	+/Nr	-	-	-	+
	dermatological	-	+/Pr	-	-	+/Pr	-	+/Pr	-	-
	orl	-	+/Pr	+/Nr	-	-	+/Pr	-	+/Pr	+
	gynecological	-	-	-	-	-	-	-	-	-
	pediatric	+	+	+	+	+	+	+	+	+
	ophthalmological	+	+	+	+	+	+	+	+	-
	medico-legal	+	+	+	+	+	+	+	+	+

Legend: +—performed, Pr—positive result, c—current fracture sites, -—non performed, Nr—negative result, p—past fracture sites.

## Data Availability

Not applicable.

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
