# Peer review of "Child Abuse: Adherence of Clinical Management to Guidelines for Diagnosis of Physical Maltreatment and Neglect in Emergency Settings"

_ijerph, 2023, doi:10.3390/ijerph20065145_

Round 1

Reviewer 1 Report

I think the purpose of the study is urgent, to highlight the need of standardized guidelines in the health services to detect child abuse. It seems critical that hospitals don't use a coroner expert in child abuse to evaluate when small children who are unable to speak on their own and tell what happened is hospitalized with serious injuries. Examination concerning possible child abuse seems to start too late to identify what caused the injuries. The fact that signs of maltreatment is often ignored in clinical evaluation seems critical. The authors also pointed out that photographical material during evaluation was missing. And a psychological evaluation of the parents/caregivers was never performed in the cases. This proves the need for more research in this field. The authors highlight that health care professionals should see the signs of child abuse earlier. They advise that there should be a standardized clinical-diagnostic protocol. The authors say that current guidelines recommend health care professionals to decide in each case whether they suspect child abuse. Then it's random whether exposed children are discovered today. The authors suggest that the child is followed over time to identify signs of child abuse. More training and knowledge should according to the authors be offered to health care professionals to detect child maltreatment.

I miss more references after 2020. There are just three, and in general few references (20). There are a lot of technical terms which the editor must review in further detail. Table 1 must also be reviewed by the editor to be sure everything is obvious. It was not all clear for me. 

I miss that the paper says some weeknesses in the study and what the authors should do in their future research in this field.

Reviewer 3 Report

The subject of the article is undoubtedly socially important and current. But the article does not meet the requirements of a scientific text. This is a report on the analysis of documentation, it contains only a description of the situation, i.e. it is only descriptive. It does not contain any explanations, understanding of the mechanisms. The authors criticize the diagnostic activities described in the medical documentation (and rightly so) and believe that they are insufficient. They also propose complementary procedures. Such a report should be sent to the relevant authorities, people who decide on procedures in medical facilities. A scientific publication should be extended with an analysis of why this is so and what are the conditions to change it. Therefore, I regret to say that, in my opinion, the article should be rejected. Figure 2 is unnecessary, it is very simple, and the same information is in the text. In the references, publications in a language other than English should have their titles translated into English.

Round 2

Reviewer 1 Report

The revised version seems ok, but I suppose there should be more about ethical considerations in this study.

Author Response

Point 1: The revised version seems ok, but I suppose there should be more about ethical considerations in this study.

Response 1: We have included some ethical ideas in the conclusions at lines 410-415. 

Reviewer 2 Report

Thank you for your patience in responding to my comments. 

Author Response

Point 1: Thank you for your patience in responding to my comments. 

Response 1: We thank you for your comments which have allowed us to improve manuscript and develop new ideas for future research.